# Alcohol consumption in the G7 countries (1960–2021). Permanent versus transitory shocks

**Luis Alberiko Gil-Alana**[1,2]*, **Gema Lopez**[3], **María Hernández-Herrera**[4]

**1** DATAI, NCID, Faculty of Economics, University of Navarra, Pamplona, Spain, **2** Department of Business, Faculty of Business Government and Law, Universidad Francisco de Vitoria, Madrid, Spain, **3** Department of Marketing, Faculty of Commerce and Tourism, Universidad Complutense de Madrid, Madrid, Spain, **4** Department of Business Administration and Management, Faculty of Economics and Business, University of Oviedo, Asturias, Spain

☉ These authors contributed equally to this work.

* alana@unav.es

**Data Availability Statement:** All relevant data are within the manuscript and its Supporting Information files.

**Funding:** MICIN-AEI-FEDER PID2020-113691RB-I00 project from 'Ministerio de Ciencia e

## Abstract

This paper analyses the degree of persistence in the level of consumption of alcohol in the Group of Seven (G7) countries by using fractional integration. The series under examination are annual sales of pure alcohol in litres per person aged 15 years and older, annually from 1960 to 2021, and we look at the influence that external shocks might have had on the series in these countries. The results indicate that only France displays a significant negative trend and thus a continuous decrease in the level of alcohol consumption. For the rest of the countries, the time trend is insignificant. Dealing with persistence, Japan is the only country that shows clear evidence of reversion to the mean. Policy recommendations are reported at the end of the manuscript.

## Introduction

Alcoholic beverages have accompanied humanity since ancient times. There is a chemical analysis that confirmed that the earliest alcoholic beverage in the world discovered so far was a mixed fermented drink of rice, honey, and hawthorn fruit and/or grape. The residues of the beverage, dated circa (ca.) 7000–6600 Before Common Era (BCE), were recovered in China [1].

In recent times, alcohol has become a salient facet of human existence, demonstrating a dichotomy of implications. On the one hand:

- Its adverse ramifications encompass an array of deleterious effects, such as the affliction of alcoholism. Alcohol consumption has a significant impact on health or addiction issues, including public health issues. It is associated with both short-term and long-term health effects [2].

- The global imperative outlined in the United Nations Agenda 2030, which entails reducing alcohol consumption as a prerequisite for achieving the Sustainable Development Goals [3].

- The economic impact of the sector makes it have enough power to influence national policy-making [4] and to resist effective alcohol policies [5].

Innovación' (MICIN), 'Agencia Estatal de Investigación' (AEI) Spain and 'Fondo Europeo de Desarrollo Regional' (FEDER). An internal project from the Universidad Francisco de Vitoria is also acknowledged.

**Competing interests:** The authors have declared that no competing interests exist.

- Its proscription within specific cultures, notably, within the context of Islamic practices [6, 7].

- While on the other hand:

- Alcohol is deeply intertwined with various cultures and societies. It plays a role in social interactions, rituals, and celebrations and also is an integral component of gastronomy [8–10].

- The industry has an economic impact, due to the production, distribution, and sale of alcohol that contribute to a nation's economy. Also, the taxes paid by the companies to the administration have an economic impact [11].

- Concurrently, alcohol is often subject to legal and regulatory frameworks, such as age restrictions and taxation. These policies can vary widely between countries. Legal statutes and norms have undergone heightened stringency in various nations, thereby exerting a palpable influence on patterns of consumption [12]. However, the imposition of stringent regulations has not invariably yielded complete abstinence, and the World Health Organization [13] recommended that additional measures must be conducted to sustain the reduction of harm attributed to alcohol.

In summary, we are interested in this topic for several reasons: 1) Health implications: Studying alcohol consumption allows us to understand its health implications, including potential benefits in moderation and risks in excess; 2) legal and regulatory frameworks: Researching alcohol consumption helps inform these legal and regulatory decisions; 3) social, behavioural and cultural aspects: Exploring alcohol consumption sheds light on its cultural significance and the social dynamics it influences; and 4) patterns of consumption and societal factors: Understanding the reasons people consume alcohol, the patterns of consumption, and the societal factors influencing these behaviours. From a marketing perspective, the analysis of the consumption of a product in the market is of special interest for predicting future behaviours and trends.

The other side of our topic is the G7 group. We have special interest in these countries because collectively they hold substantial economic, political, and policy influence on the global stage. Their data provide valuable information for understanding worldwide economic trends and performance, trade dynamics, and geopolitical developments. G7 nations often play a pivotal role in shaping international policies and agreements.

Table 1 shows the primacy of these countries and the positive numbers that they manage. They are role models in terms of living standards and economic performance.

The remainder of the paper is structured as follows: Section 1 introduces the topic and provides a review of the relevant literature on consumption of alcohol and about the G7 countries. Section 2 outlines the methodology employed, detailing the fractional integration framework, and explains the data used across the seven countries. Section 3 presents the empirical results, analysing the persistence properties of alcohol consumption and identifying potential country-specific variations, which are discussed in Section 4. Lastly, Section 5 concludes the article, summarising the key findings and offering suggestions about policy implications across these seven nations.

## Literature review

It is interesting to review the literature about the reasons for alcohol consumption to understand the increase or decrease in it. Thus, for example, Gual and Colom [15] observed a decline in alcohol consumption in the traditional wine-producing countries of Southern Europe. In

**Table 1. Data for the G7 countries in 2022.**

| Country | SMI Salary | CPI* | GDP growth (annual %) | Standard of living** | Economy ranking*** |
|---|---|---|---|---|---|
| Canada | €1,757.1 | 3.8 | 3.4 | Very good | <10 |
| France | €1,747.2 | 4.9 | 2.5 | Good | <10 |
| Germany | €1,997 | 4.5 | 1.8 | One of the best | <10 |
| Italy | – | 5.3 | 3.7 | Good | <10 |
| Japan | €1,246.7 | 3.1 | 1 | Good | 3 |
| United Kingdom (UK) | €1.929.2 | 6.7 | 4.3 | Good | <10 |
| United States (US) | €1,178.2 | 3.7 | 1.9 | Good | 1 |

* Interannual in September 2023

** According to the GDP, the standard of living in relation to the rest of the 196 countries in the GDP per capita ranking

*** By volume of GDP. Source: Own elaboration. Data from: Expansión [14].

their conclusions, they found a trend toward the homogenisation of alcohol consumption patterns. Factors such as marketing strategies, public health policies, price and tax developments, European Union agricultural policies, an increasing public awareness of alcohol's toxicity, and the competition from non-alcoholic beverages were suggested as partial explanations for these observed changes. Vuik and Cheatley [16] found that, on a general basis across Organisation for Economic Co-operation and Development (OECD) nations, individuals with higher income tend to have a higher likelihood of engaging in weekly drinking and binge drinking compared to those in lower income brackets. In nearly all countries that they analysed (OECD and Group of Twenty -G20-), individuals who have obtained tertiary or university education are more inclined to engage in weekly drinking, with this effect being particularly pronounced among women. A prevailing trend over the past two decades reveals that alcohol has become more economically accessible in nearly all OECD countries, primarily driven by an increase in real income. Additionally, the initiation of alcohol consumption at an early age is linked to a continued pattern of drinking in later life.

Specifically, for the G7 countries, Kairouz et al. [17] found in Canada that the combination of motivations for drinking and the environment in which drinking occurs collectively affects the amount of alcohol consumed. Additionally, the motivations behind drinking vary depending on the specific situation, as students have different reasons for drinking in different contexts. Thakore et al. [18] discovered that the consumption of alcohol and tobacco among medical students at the University of Calgary is lower than that of a similar demographic. Nevertheless, a significant portion of students exhibit characteristics that put them at risk of alcohol abuse, as indicated by the CAGE questionnaire. Since the sample is based on medical students, it might be inferred that the information related to the consequences of alcohol consumption persuades them to consume it.

For France, De Chazeron et al. [19] revealed, in the first decade of the 2000s, that a significant number of pregnant women continue to consume alcohol, especially during episodes of binge drinking. Andler et al. [20] described that one-quarter of the adults reported alcohol consumption exceeding low-risk guidelines, with women consuming less than men. Furthermore, younger individuals drank less frequently, but with greater intensity compared to older ones. Initial alcohol consumption patterns were observed during middle school, and some regular usage was already established in high school. In this review, we have reflected the literature most related to our topic, although we have not found previous analyses that explain the linear decline in alcohol consumption in France that we observe from 23.2 in 1970 to 10.5 in 2021.

For Germany, Donath et al. [21] highlighted that binge drinking is a prevalent issue among German adolescents. It is apparent that adolescents residing in rural areas have fewer options

for participating in engaging leisure activities compared to their urban counterparts. Wolf et al. [22] examined data from 1997–99 and 2008–11 that indicated shifts in alcohol consumption trends among the elderly demographic. Moderate alcohol consumption increased significantly.

For Italy, Allamani et al. [23] found that sociodemographic and economic factors, particularly urbanisation, were conclusively linked to the decline in spirits consumption. On the other hand, alcohol control policies did not show any effect on the decrease in alcohol consumption, as there was no alcohol control policy in Italy between 1960 and 1987. Some policies introduced since 1988 (such as blood alcohol concentration limits and restrictions on alcohol sales during mass events) may have contributed to reducing or sustaining the ongoing decline. Laghi et al. [24] showed that binge drinking is a common behaviour among Italian adolescents, with their primary motivations for drinking being to amplify the perceived positive effects of alcohol, conform to social norms within their peer groups, and cope with personal challenges. It was observed that boys engage in binge drinking more frequently than girls.

For Japan, Nagoshi et al. [25] compared university students in US vs Japan in their alcohol use finding that while the frequencies of drinkers versus abstainers did not differ between the two samples, American students initiated regular alcohol consumption at a significantly earlier age, currently consumed more alcohol, held higher alcohol outcome expectancies regarding emotional responses, and endorsed more celebratory motives for drinking than their Japanese counterparts. However, American students had lower expectations of blushing and perceived lower drinking norms. Higuchi et al. [26] found that the diversification of the drinking population has advanced swiftly, particularly among women, among whom alcohol consumption has increased significantly. Alcohol policies and prevention programs have not been developed to a level that can effectively manage health issues or even fatal accidents. The high availability of alcoholic beverages, including the absence of sales and advertising restrictions and reduced prices, is emphasised.

For the US, the majority of the literature relates alcohol consumption to its impact on consumer mortality, highlighting it as a major concern. We found very few studies on the historical consumption patterns in this country. We reference the following: Grossman et al. [27] summarised three reasons for increasing drinking during the COVID-19 pandemic: Increased alcohol availability, boredom and increased stress. Additionally, Ritchie and Roser [28] present data on alcohol consumption in the US, showing an increase from 2000 to 2019, consistent with the data presented in this analysis.

For the UK, Smith and Foxcroft [29] explained the possible reasons for the increase in alcohol consumption in the different groups:

- For women, because they have achieved greater equality in various aspects of life, resulting in increased disposable income and more women delaying marriage or getting divorced and thus having fewer family responsibilities. The societal acceptance of alcohol consumption by women has grown, and alcohol is now more readily available in supermarkets and stores, with drinking environments becoming less exclusive to men. There is also a noticeable rise in advertising and promotional efforts targeted towards women.

- For 16 to 24 years old, because the marketing and promotional campaigns are designed to attract a younger demographic. The night-time economy has expanded, offering increased access to more affordable alcohol and greater opportunities for young people to consume alcohol in larger quantities. Young individuals in the UK hold heightened positive expectations regarding the effects of alcohol. The adolescent phase, typically associated with riskier behaviours, has extended, with more young people continuing to live at home into their twenties.

- For middle and older age groups, because of increased wealth, earlier retirement resulting in more leisure time, better health, and a youthful perspective. Additionally, alcohol has become more cost-effective. Factors such as boredom, loneliness, and challenges associated with adjusting to changing life roles, such as caregiving for older family members or dealing with empty nest syndrome, also play a role. Alcohol education has predominantly targeted binge and problem drinking in younger age groups, potentially leaving older individuals less informed about recommended limits and associated health risks.

We would like to highlight that in the reviewed literature, we have not found any article or book that analyses the issue of alcohol consumption from a time series perspective, which makes this analysis on the persistence of alcohol consumption in G7 countries using fractional integration potentially useful in the field. Also, no related literature was found corresponding to the research results for the analysed countries.

## Materials and methods

### Methodology

The model under investigation is quite simple and based on the following two equations:

$$y_t = \alpha + \beta t + x_t, \quad (1 - L)^d x_t = u_t, \quad t = 1, 2, \dots, \tag{1}$$

where $y_t$ refers to the time series under examination, i.e., the alcohol consumption in the G7 countries. The first equation refers to the deterministic part of the model that is described in terms of a linear time trend as is standard in the time series literature [30, 31]. Employing non-linear deterministic models like those based on Chebyshev polynomials in time [32] produced qualitatively the same results as those reported in the following section. The second equation describes the stochastic part of the model, which is a fractionally integrated or an I(d) model where d can be any real value, and thus, permitting fractional numbers. L is the lag operator, i.e., $Lx_t = x_{t-1}$, and $u_t$ is a short memory or I(0) process that will be expressed under the assumption of no autocorrelation, so that $u_t$ is a white noise process with zero mean a constant variance.

Therefore, there are two relevant parameters in this model: β, representing the time trend in the data, and d, that is a measure of persistence. To see this, note that the polynomial in L in the second equality in (1) may be expanding for any real value d as:

$$(1 - L)^d = \sum_{j=0}^{\infty} \binom{d}{j} (-1)^j L^j = 1 - dL + \frac{d(d-1)}{2} L^2 - \cdots$$

and thus, the second equality in (1) can be expressed as

$$x_t = d x_{t-1} - \frac{d(d-1)}{2} x_{t-2} + \dots + u_t.$$

In this context, if d is a fractional value, $x_t$ will be a function of all its past history represented in terms of an infinite AR process.

In the empirical application carried out in the following section, we employ a version of a testing procedure developed in Robinson [33], which tests the null hypothesis $H_o$: $d = d_o$ in (1) where $d_o$ is a given real value. This method has numerous advantages with respect to other approaches. First, it relies on a standard normal asymptotic distribution, and this holds independently of the inclusion of deterministic terms like those based on the linear time trend as in (1). Moreover, $d_o$ can be any value, and thus we do not have to restrict ourselves to the stationary range (i.e., $d_o < 0.5$); finally, it has been proved to be the most efficient method in the

Pitman sense against local departures from the null. Its functional form can be found in any of the numerous applications that use this approach [34]. Though not reported, employing alternative fractionally integrated parametric and semiparametric approaches produced essentially the same results.

## Data

The analysed data is from OECD. This is an international organisation that works to build better policies for better lives. Their goal is to shape policies that foster prosperity, equality, opportunity and well-being for all [35]. They publish data about different topics among which we can find alcohol consumption [36]. The topic analysed is alcohol consumption as annual sales of pure alcohol in litres per person aged 15 years and older. For the OECD this topic is extremely important due to association with harmful health and social consequences, including death and disability. Its observation and analysis should help in its objective of improving the common well-being. We choose the G7 as our focus of research: Canada, France, Germany, Italy, Japan, UK and US. The observed dates were between 1960 to 2021.

As we can observe in Fig 1, not all the countries have had the same evolution: France and Italy have experimented a higher decrease in consumption. Germany increased for a few years until 1976, before a linear decrease until now, finishing with a very similar number to the beginning. Canada displays a linear pattern with ups and downs across the years. Japan, the UK and the US display a linear increase during the observed period.

Table 2 displays some descriptive statistics. The maximum level of consumption varies across similar observed dates. France has the record with 23 litres/capita in contrast with Japan whose maximum is 9.2 litres. The minimum consumption is 1.9 litres in Japan followed by Canada, Italy, UK and US at around 7 litres. On average, France boasts the highest value at 15.8326, while Japan records the lowest at 7.3258. Finally, on the standard deviation, Italy has the highest number with 4.6278 while US has the lowest one with 0.7313. In summary the most extreme countries in terms of alcohol consumption numbers are France and Japan.

The data presented shows trends and the history of alcohol consumption in these countries but does not directly indicate whether, in the event of a drastic change in the figures, they would recover as a temporary change or whether the historical series would be affected in subsequent years. For this purpose, we apply the fractional analysis explained in the methodology.

## Empirical results

Table 3 displays the results in terms of the estimated values of d for the original data. We report the values of the differencing parameter, i.e., d in Eq (1) along with their corresponding confidence intervals at the 95% level, under three different assumptions:

i.  imposing that $\alpha = \beta = 0$ a priori, i.e., we do not include either a constant nor a time trend (results reported in the second column of the table),

ii.  imposing that $\beta = 0$ a priori, i.e., we include an intercept but not a time trend (results reported across the third column in the table), and

iii.  with a constant and a linear time trend, so $\alpha$ and $\beta$ are estimated along with d (column 4).

We mark in bold in Table 3 the selected specification for each series, this selection being made based on the significance of these deterministic terms. Thus, if both $\alpha$ and $\beta$ are statistically significant different from zero, we choose the model in column 4; however, if $\beta$ is insignificant we choose those values reported in column 3 and based on an intercept. If both

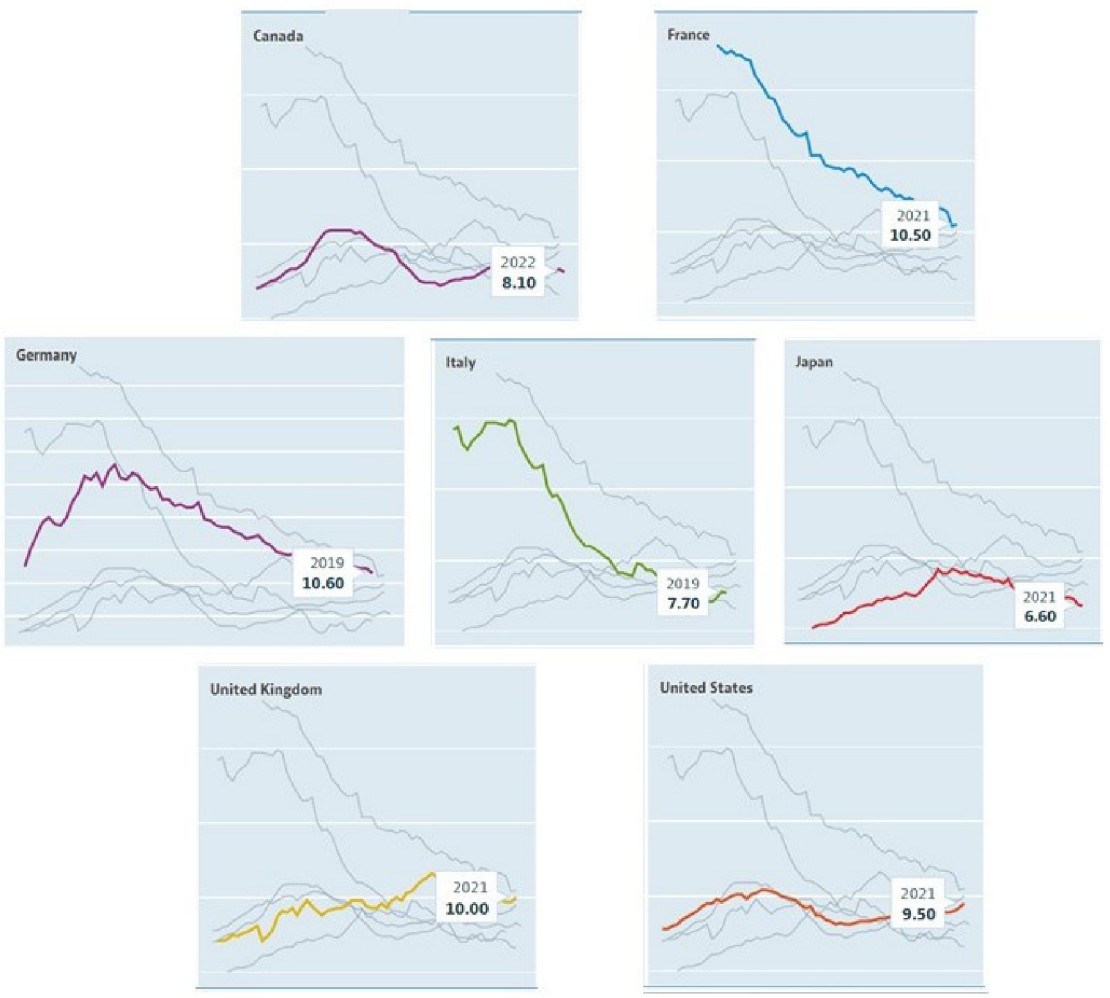

**Fig 1. Time series plots: Alcohol consumption.** Source: OECD, 2023.

coefficients are insignificant, we choose those in column 2 with no deterministic terms. Tables 4 and 5 reproduce Tables 3 and 6 but for the log-transformed values, and though quantitatively there are some differences in the values of d, qualitatively the results are very similar. We discuss the results reported across these tables in the following section.

**Table 2. Descriptive statistics of alcohol consumption in G7 countries (in litres/capita).**

| Series | Dates | Max. | Min. | Mean | Std. Dev.$<$ |
|---|---|---|---|---|---|
| Canada | 1960–2021 | 10.9 | 7 | 8.6225 | 1.1903 |
| France | 1970–2021 | 23.2 | 10.4 | 15.8326 | 3.9147 |
| Germany | 1961–2019 | 17.2 | 10.6 | 13.5830 | 1.9718 |
| Italy | 1962–2019 | 19.9 | 7 | 12.5862 | 4.6278 |
| Japan | 1964–2021 | 9.2 | 1.9 | 7.3258 | 1.3024 |
| UK | 1960–2021 | 11.6 | 7.1 | 9.3409 | 1.1287 |
| US | 1960–2021 | 10.4 | 7.8 | 9.0274 | 0.7313 |

**Table 3. Estimates of the differencing parameter, d, in Eq (1). Original data.**

| Country | No terms | An intercept | An intercept and a linear time trend |
|---|---|---|---|
| **Canada** | 1.06 (0.92, 1.26) | **1.65 (1.49, 1.88)** | 1.63 (1.47, 1.86) |
| **France** | 0.94 (0.77, 1.19) | 0.92 (0.79, 1.13) | **0.94 (0.82, 1.10)** |
| **Germany** | 1.06 (0.92, 1.27) | **1.15 (1.03, 1.33)** | 1.15 (1.03, 1.31) |
| **Italy** | 0.96 (0.82, 1.16) | **1.23 (1.05, 1.56)** | 1.24 (1.05, 1.57) |
| **Japan** | 0.58 (0.40, 0.77)* | **0.44 (0.33, 0.57)*** | 0.44 (0.33, 0.59) * |
| **UK** | 0.97 (0.80, 1.21) | **1.01 (0.83, 1.27)** | 1.01 (0.85, 1.26) |
| **US** | 1.02 (0.88, 1.26) | **1.27 (1.16, 1.42)** | 1.26 (1.15, 1.40) |

The values reported in the tables are the estimates of the differencing parameter d. Those in parenthesis are their associated 95% confidence intervals. In parenthesis the selected specification for each series

* Evidence of mean reversion at the 95% level. In bold, the selected deterministic approach for each country.

## Discussion

The first thing we observe in Table 3 is that all the estimates of the integration parameter d are significantly higher than 0 indicating evidence of fractional integration. We also observe that the time trend is only required in the case of France. For the rest of the countries, the intercept is the only deterministic term required. Focusing on the specific coefficients for each country, (in Table 6), we see that the time trend coefficient is negative in the case of France and looking at the orders of integration of the series, we see that the seven countries can be grouped into three categories: 1) There is a single country, Japan, with evidence of mean reversion, (i.e., with a value of d significantly below 1). The estimated coefficient is 0.44. Thus, shocks in this series will clearly have a transitory effect, disappearing by themselves in the long run; 2) there are two countries where the null hypothesis of a unit root, i.e., d = 1 cannot be rejected. They are France (d = 0.94) and UK (d = 1.01). Note that in the case of France, d is below 1 but its confidence interval includes 1 and also values higher than 1; 3) for the rest of the countries, the estimates of d are significantly above 1. These values are 1.15 (Germany); 1.23 (Italy), 1.27 (US) and 1.65 (Canada). Thus, only Japan presents evidence of reversion to the mean and the North American countries display the highest degrees of persistence.

France is the only country displaying a significant (negative) trend; a difference comes in this country, France, displaying now with Japan evidence of mean reversion, with an estimate of d of 0.27 for Japan and much higher, 0.74 for France; for the UK, the unit root null cannot

**Table 4. Estimates of the differencing parameter, d, in Eq (1). Logged data.**

| Country | No terms | An intercept | An intercept and a linear time trend |
|---|---|---|---|
| **Canada** | 0.98 (0.84, 1.20) | **1.65 (1.49, 1.89)** | 1.61 (1.46, 1.85) |
| **France** | 0.93 (0.76, 1.19) | 0.79 (0.71, 0.95)* | **0.75 (0.62, 0.95)*** |
| **Germany** | 0.99 (0.83, 1.21) | **1.18 (1.06, 1.37)** | 1.18 (1.06, 1.34) |
| **Italy** | 0.96 (0.81, 1.18) | **1.18 (1.00, 1.49)** | 1.19 (1.00, 1.49) |
| **Japan** | 0.65 (0.48, 0.85)* | **0.27 (0.15, 0.43)*** | 0.26 (0.14, 0.43)* |
| **UK** | 0.95 (0.79, 1.19) | **0.97 (0.78, 1.24)** | 0.98 (0.81, 1.23) |
| **US** | 0.98 (0.82, 1.19) | **1.28 (1.17, 1.42)** | 1.26 (1.16, 1.40) |

The values reported in the tables are the estimates of the differencing parameter d. Those in parenthesis are their associated 95% confidence intervals. In parenthesis the selected specification for each series

* Evidence of mean reversion at the 95% level. In bold, the selected deterministic approach for each country.

**Table 5. Estimated coefficients based on the selected model in Table 4.** Logged data.

| Country | d (95% band) | Intercept (tv) | Time trend (tv) |
|---|---|---|---|
| Canada | 1.65 (1.49, 1.89) | 1.937 (131.14) | —— |
| France | 0.75 (0.62, 0.95)* | 3.161 (143.64) | -0.015 (-11.48) |
| Germany | 1.18 (1.06, 1.37) | 2.376 (81.80) | —— |
| Italy | 1.18 (1.00, 1.49) | 2.976 (84.64) | —— |
| Japan | 0.27 (0.15, 0.43)* | 1.925 (27.71) | —— |
| UK | 0.97 (0.78, 1.24) | 1.961 (54.68) | —— |
| US | 1.28 (1.17, 1.42) | 2.049 (123.01) | —— |

The values in parenthesis in columns 3 and 4 are the associated t-values of the estimated deterministic terms
* Evidence of mean reversion at the 95% level.

be rejected (0.97), and the estimates of d are much higher than 1 in the remaining countries: Germany and Italy (with d = 1.18 in the two countries), US (1.28) and Canada (1.65).

In relation to specific challenges and preventive measures in each country:

- France has a very strong, deep-rooted culture of alcohol consumption, especially in wine, but alcohol consumption has steadily decreased since the 1950s, primarily in connection with the decline in wine consumption, marked by the abandonment of "table wine" in favour of higher-quality wines consumed in smaller quantities. However, in terms of policy, France's efforts in public health awareness have effectively contributed to reducing alcohol consumption and it must therefore continue with the measures taken and monitor the figures in case the downward trend changes. While France remains one of the countries with higher alcohol consumption in the European Union, it no longer holds the top position in the ranking [37]. The trend here indicates that shocks will have transitory effects, meaning the consumption pattern will recover without significant intervention.

- Japan shows clear evidence of mean reversion, meaning shocks to alcohol consumption tend to be temporary. A significant factor has been the rise in alcohol consumption among women, particularly in the last few decades, accompanied by limited policy restrictions on alcohol sales and advertising. The high availability of alcohol and relatively low prices contribute to this challenge. In contrast to France, Japan's policy measures are insufficient to mitigate these risks effectively, requiring stronger regulatory frameworks targeting female consumption and low prices, with the possibility of increasing alcohol taxes.

**Table 6. Estimated coefficients based on the selected model in Table 3.** Original data.

| Country | d (95% band) | Intercept (tv) | Time trend (tv) |
|---|---|---|---|
| Canada | 0.44 (0.33, 0.57) | 6.940 (53.53) | —— |
| France | 1.01 (0.83, 1.27) | 23.442 (65.12) | -0.249 (-6.17) |
| Germany | 1.27 (1.16, 1.42) | 10.777 (25.84) | —— |
| Italy | 1.23 (1.05, 1.56) | 19.642 (45.61) | —— |
| Japan | 0.44 (0.33, 0.57)* | 6.483 (11.70) | —— |
| UK | 1.01 (0.83, 1.27) | 7.096 (22.24) | —— |
| US | 1.27 (1.16, 1.42) | 7.763 (50.66) | —— |

The values in parenthesis in columns 3 and 4 are the associated t-values of the estimated deterministic terms
* Evidence of mean reversion at the 95% level.

- The United States and Canada display the highest levels of persistence, indicating that any shock (whether an increase or decrease in alcohol consumption) is likely to have lasting effects. Public health campaigns, while present, are insufficient to combat the high persistence in alcohol consumption trends. There is a need for more robust regulatory and taxation measures, along with more aggressive health awareness campaigns to mitigate the long-term risks.

- Germany and Italy present medium-to-high persistence in shocks. The reduction in alcohol consumption seems more closely linked to sociodemographic and economic factors such as urbanisation. In both countries, persistence in consumption remains relatively high, suggesting that social marketing and government campaigns could help bring about the social behavioural change needed to curb alcohol consumption figures.

- In the UK, our results show that shocks in alcohol consumption tend to be temporary. Social and economic factors have played a major role in shaping alcohol consumption patterns, especially among women and young adults. Increased purchasing power and changes in social norms have led to higher consumption levels, above all in urban areas. Like in the US and Canada, the UK presents high persistence in alcohol consumption, meaning that stronger measures are necessary to curb this trend. Policy changes targeting specific demographic groups, such as young adults and women, could help mitigate long-term consumption trends.

While the trends in alcohol consumption vary across the G7 nations, the countries exhibiting the highest levels of persistence will require stronger and more targeted policy interventions to manage shocks to alcohol consumption effectively. Countries like France and Japan, where mean reversion is more evident, may experience temporary disruptions in consumption but will likely return to their established trends with minimal intervention.

## Concluding comments

These nations represent well-developed economies, and as a result, they exhibit higher consumption levels in the upper tiers of Maslow's hierarchy, such as social and self-confidence. Alcohol is also ingrained in their culture, and it is not perceived as having negative short-term effects [38].

But the data shows us that in several cases consumption has been reduced due to different reasons:

- Increased health awareness: Greater awareness of the health risks associated with excessive alcohol consumption, such as liver diseases, cancer, and cardiovascular issues, has motivated many people to decrease their alcohol intake. Also, the increasing concern for overall wellness and health has led to an interest in reducing alcohol consumption as part of a healthier lifestyle [13, 39].

- Changes in social norms: There has been a shift in social norms surrounding alcohol consumption in many developed countries, making excessive alcohol consumption less socially acceptable and leading to a reduction in consumption [40].

- Prevention and education campaigns: In line with the previous reasons, we find that dissemination is essential. Prevention and education campaigns about the risks of alcohol, conducted by governments and health organizations, have played a role in informing the public about the dangers of excessive alcohol consumption. While it is essential to be mindful of the impact on consumption, it is not highly significant or quick [41].

- Stricter regulations: The implementation of stricter laws regarding consumption and advertising of alcohol has had an impact on reducing its use [42]. For example, to reduce alcohol consumption, an increase in taxes would be more effective than sales restrictions [43].

  Also, considering the country-specific findings, we provide a set of policy recommendations that can address the patterns observed in the data:

- Focus on persistence and shock management: Countries with medium-to-high persistence (such as Canada, US, Germany, Italy) will have to maintain policies oriented towards addressing the long-run trends. In these countries, shocks to alcohol consumption have permanent effects, suggesting that sustained policy interventions are required to change long-term behaviours. Higher taxes on alcohol, structuring of the market, and closely regulating advertising would make the biggest impact on reducing consumption. These interventions should focus not only on the price mechanisms but also on public health messaging that highlights the risks of sustained alcohol use over time [44].

- Youth and vulnerable groups: Younger populations and some at risk groups (for example, women in Japan and the UK; adolescents in Italy) are drinking more alcohol, with indices moving in an adverse direction across the G7. Preventive interventions might be better addressed through educational campaigns among the youngest drinkers, stricter enforcement of age limits, and encouraging alcohol-free alternatives [45]. School-based interventions and social marketing campaigns can be key in shaping the attitudes and behaviours of young adults before harmful drinking patterns become established.

- Health Literacy and Tendency to Accept Societal Norms: Misconceptions lead individuals to adopt negative behaviours, such as consuming higher amounts of alcohol and other substances [46]. It is essential to mitigate the occurrence of episodic heavy drinking by implementing targeted prevention strategies [47]. One of the most significant factors reducing alcohol consumption in countries like France has been the shift in social norms towards healthier lifestyles. As a result, we should be working to shape the same sensibilities among our G7 neighbours with public health campaigns promoting moderation and highlighting the inherent risks of consuming one more drink. Governments need to use social media and modern communication to keep altering the image of drinking in societies where it is culturally embedded.

- Regulatory and fiscal policies: Taxation and regulation are well recognised as the most effective policy measures to lower alcohol use [48]. Moderate impacts can also be expected from stricter regulation of availability, such as restrictions on hours of sale or advertising [49] and higher alcohol taxes to discourage excessive consumption. Some of these policies, however, will have to be designed very prudently vis-a-vis the socioeconomic demographics of each country, in order to avoid unintended consequences like black-market sales or hitting lower-income groups disproportionately.

- Monitoring and evaluation: Adopting a flexible approach to regulating alcohol. Countries should watch trends in the consumption of these substances and change any relevant regulations depending on the prevailing tendencies. Without policy evaluation mechanisms, it would be difficult to determine to what extent these brief interventions are proving effective in reducing alcohol consumption and impacting public health. These adjustments should be data-driven and longitudinal studies of public health data.

  Although socio-cultural challenges related to alcohol consumption are country specific (see discussion), reducing total alcohol consumption and related harm requires an integrated,

comprehensive approach that includes fiscal, regulatory, educational and health strategies. In countries with high persistence, long-term policy consistency is critical, while in countries with evidence of mean reversion (such as Japan and France), policies should aim to reinforce and sustain positive trends.

## Supporting information

**S1 File. Database.**
(PDF)

## Acknowledgments

Comments from the Editor and two anonymous reviewers are gratefully acknowledged.

## Author Contributions

**Formal analysis:** Luis Alberiko Gil-Alana.

**Funding acquisition:** Luis Alberiko Gil-Alana.

**Methodology:** Luis Alberiko Gil-Alana.

**Writing – original draft:** Luis Alberiko Gil-Alana, Gema Lopez, María Hernández-Herrera.

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
