## [Decision Letter · Decision Letter 0]

3 Sep 2024

PONE-D-24-04353Alcohol Consumption in the G7 Countries (1960-2021). Permanent versus Transitory Shocks.PLOS ONE

Dear Dr. Gil-Alana,

Thank you for submitting your manuscript to PLOS ONE. After careful consideration, we feel that it has merit but does not fully meet PLOS ONE’s publication criteria as it currently stands. Therefore, we invite you to submit a revised version of the manuscript that addresses the points raised during the review process.

Please submit your revised manuscript by Oct 18 2024 11:59PM. If you will need more time than this to complete your revisions, please reply to this message or contact the journal office at plosone@plos.org. Please include the following items when submitting your revised manuscript:A rebuttal letter that responds to each point raised by the academic editor and reviewer(s). You should upload this letter as a separate file labeled 'Response to Reviewers'.A marked-up copy of your manuscript that highlights changes made to the original version. You should upload this as a separate file labeled 'Revised Manuscript with Track Changes'.An unmarked version of your revised paper without tracked changes. You should upload this as a separate file labeled 'Manuscript'.

We look forward to receiving your revised manuscript.

Kind regards,

Ricardas Radisauskas

Academic Editor

PLOS ONE

Journal Requirements:

"MICIN-AEI-FEDER PID2020-113691RB-I00 project from ‘Ministerio de Ciencia e Innovación’ (MICIN), `Agencia Estatal de Investigación' (AEI) Spain and `Fondo Europeo de Desarrollo Regional' (FEDER). An internal project from the Universidad Francisco de Vitoria is also acknowledged."

Additional Editor Comments:

Dear manuscript authors,

Thank you for your submitted manuscript.

Currently, the manuscript still has some shortcomings, which the authors of the manuscript must correct based on the comments provided by the reviewers.

Reviewers' comments:

Reviewer's Responses to Questions

**Comments to the Author**

1. Is the manuscript technically sound, and do the data support the conclusions?

Reviewer #1: Partly

Reviewer #2: Yes

2. Has the statistical analysis been performed appropriately and rigorously? 

Reviewer #1: I Don't Know

Reviewer #2: Yes

3. Have the authors made all data underlying the findings in their manuscript fully available?

Reviewer #1: Yes

Reviewer #2: Yes

4. Is the manuscript presented in an intelligible fashion and written in standard English?

Reviewer #1: Yes

Reviewer #2: Yes

5. Review Comments to the Author

Reviewer #1: Thank you for the opportunity to review this research work.

The authors have analyzed the alterations over time in the level of consumption of alcohol in the G7 countries by using a mathematical modelling i.e., fractional integration. The data used for the analyses include sales of pure alcohol in liters per person aged 15 years and older, annually from 1960 to 2021 for each country. The influence of external shocks on the sales is predicted based on the applied model. The authors have shown a significant negative time trend for the level of alcohol consumption in France and a tendency to reversion to mean for Japan, but no significant findings for the other countries. They refer to policy recommendations aiming to reduce alcohol consumption in their conclusions.

While the research work seems interesting and creatively designed, there are some limitations which should be addressed to improve the paper.

Although the findings are discussed in the context of the previous literature, one cannot see that the related literature reflect the accurate observations based on the obtained results for the investigated countries. There are some discrepancies between the contains of the given text used for literature review for each country and the concluding remarks. For example, under literature review it is stated that alcohol consumption has increased significantly in Japan and the effective policy measures are absent, however, under the empirical results it is stated that shocks in this series will clearly have a transitory effect, disappearing by themselves in the long run. This seems counterintuitive.

The conclusions on a negative time trend in alcohol consumption in France do not address the specific concerns described under literature review regarding consumption patterns.

The background literature provided for USA is too short to support the implications given in the conclusion comments regarding concerns on the permanency of shocks and the need for strong measures to recover their original trends. More detailed information could be useful.

The concrete findings from each country are expected to be discussed in the light of existing challenges and the risk factors or preventing measures which are specific for the country.

Although the data analyses and applied model seems sound and well-explained, the methodological limitations of such mathematical modelling are not included in the paper.

The manuscript is well-written; however, it is organized in a different way than it usually must appear (does not include discussion, strengths and limitations).

The paper would be improved by relating the proper background information from each country to the findings and discussing the results in the light of the specific situations for the country. Thus, the implications and conclusion comments should be balanced and adjusted accordingly, rather than reporting some general and already known policy recommendations.

Reviewer #2: The manuscript covers analysis of of alcohol consumption trends in Gt countries during the very long period (1960-2021). To my opinion the manuscript is written by authors very well using using simple way of presentation of the data and understandable for each reader. I just have few technical remarks for the authors of the manuscript:

1. Each abbreviation should be described at first use in the text (for example OECD). Such abbreviation first time used in page 5 of the manuscript but described only in page 10 of the manuscript. Please describe the meaning of the abbreviations SMI, CPI, and GDP used in Table 1 (for example, in the text at the bottom of the Table1.

2. Please modify title of the Table 2. For example, Descriptive statistics of alcohol consumption in G7 countries (in litres/capita) or so on.

3. Please indicate and explain what mean bolded values in Tables 3, 4 and 5.

My recommendation is accept the manuscript after minor revision.

6. PLOS authors have the option to publish the peer review history of their article (what does this mean?). If published, this will include your full peer review and any attached files.

Reviewer #1: No

Reviewer #2: **Yes: **Abdonas Tamosiunas

---

## [Author Response · Author response to Decision Letter 0]

18 Oct 2024

Dear Editor,

Many thanks for giving us the opportunity to revise and resubmit our paper entitled "Alcohol Consumption in the G7 Countries (1960-2021). Permanent versus Transitory Shocks" in PLOS ONE.

Following your recommendation and the comments of the reviewers, we have made several changes in the paper that are summarised below.

I look forward to your response on this revised version.

Yours sincerely,

Prof. Luis A. Gil-Alana

P.S. The funders had no role in study design, data collection and analysis, decision to publish, or preparation of the manuscript.

Comments from the Editor:

Reply: We have done so in the updated version. 

2.) Thank you for stating the following financial disclosure: 

"MICIN-AEI-FEDER PID2020-113691RB-I00 project from ‘Ministerio de Ciencia e Innovación’ (MICIN), `Agencia Estatal de Investigación' (AEI) Spain and `Fondo Europeo de Desarrollo Regional' (FEDER). An internal project from the Universidad Francisco de Vitoria is also acknowledged."

 Reply: Done. We have included the sentence "The funders had no role in study design, data collection and analysis, decision to publish, or preparation of the manuscript." in the new cover letter.

Additional Editor Comments:

Thank you for your submitted manuscript.

Currently, the manuscript still has some shortcomings, which the authors of the manuscript must correct based on the comments provided by the reviewers.

Reviewers' comments:

Reviewer's Responses to Questions

Comments to the Author

1.) Is the manuscript technically sound, and do the data support the conclusions?

Reviewer #1: Partly

See reply below.

Reviewer #2: Yes

Thank you.

2.) Has the statistical analysis been performed appropriately and rigorously? 

Reviewer #1: I Don't Know

Reviewer #2: Yes

Thank you.

3.) Have the authors made all data underlying the findings in their manuscript fully available?

Reviewer #1: Yes

Reviewer #2: Yes

Thank you.

4.) Is the manuscript presented in an intelligible fashion and written in standard English?

Reviewer #1: Yes

Reviewer #2: Yes

Thank you.

Comments from Reviewer #1: 

Thank you for the opportunity to review this research work.

The authors have analyzed the alterations over time in the level of consumption of alcohol in the G7 countries by using a mathematical modelling i.e., fractional integration. The data used for the analyses include sales of pure alcohol in liters per person aged 15 years and older, annually from 1960 to 2021 for each country. The influence of external shocks on the sales is predicted based on the applied model. The authors have shown a significant negative time trend for the level of alcohol consumption in France and a tendency to reversion to mean for Japan, but no significant findings for the other countries. They refer to policy recommendations aiming to reduce alcohol consumption in their conclusions.

1.) While the research work seems interesting and creatively designed, there are some limitations which should be addressed to improve the paper. Although the findings are discussed in the context of the previous literature, one cannot see that the related literature reflect the accurate observations based on the obtained results for the investigated countries. 

Reply: Thanks. The comment is timely and correct. We have conducted a new literature search based on the results obtained for the investigated countries, and as in the original analysis, no relevant studies were found. Consequently, no specific literature could be cited. The research topics are often specific, focusing on specific consumer groups, shorter periods, countermeasures, or water consumption rather than alcohol. No studies with a time span as extensive as ours or directly related to our findings were identified. We have included the following statement in this regard: 

" … Also, no related literature was found corresponding to the research results for the analysed countries. …". (Page 9).

2.) There are some discrepancies between the contains of the given text used for literature review for each country and the concluding remarks. For example, under literature review it is stated that alcohol consumption has increased significantly in Japan and the effective policy measures are absent, however, under the empirical results it is stated that shocks in this series will clearly have a transitory effect, disappearing by themselves in the long run. This seems counterintuitive.

Reply: Thanks. As previously mentioned, there is no literature directly related to the results, so the studies cited in that section support or partially explain the results but do not fully account for them. In the case mentioned in the comment, Japan, there is an increase in alcohol consumption figures from 1963 (5) to 1996 (9), followed by a decrease until 2021 (6.6), which aligns with the cited literature and the observed increase in consumption. Regarding the comment: “However, under the empirical results, it is stated that shocks in this series will clearly have a transitory effect, disappearing by themselves in the long run. This seems contradictory,” this is not the case. The conducted analysis determines whether, in the event of a drastic change in the figures, they would recover as a temporary change or, conversely, whether the historical series would be affected in the subsequent years. This result is independent of whether the data has increased or decreased, as it considers how and over which periods these changes occurred. Therefore, initial observations of the data might suggest a conclusion or expectation that may or may not hold when applying fractional integration analysis. To clarify this aspect, we have added the following content: 

“ … The data presented shows trends and the history of alcohol consumption in these countries but do not directly indicate whether, in the event of a drastic change in the figures, they would recover as a temporary change or whether the historical series would be affected in subsequent years. For this purpose, we apply the fractional integration analysis explained in the methodology. … ”. (Page 11).

3.) The conclusions on a negative time trend in alcohol consumption in France do not address the specific concerns described under literature review regarding consumption patterns.

Reply: We understand the comment. The literature found on the topic focuses on specific periods or groups, such as “Trends in alcohol consumption in Spain, Portugal, France, and Italy from the 1950s until the 1980s” or “Changes in smoking and alcohol consumption during COVID-19-related lockdown: a cross-sectional study in France”, concentrating on a specific period of our data, or “Participation in sports and alcohol consumption among French adolescents”, which targets a specific audience. Therefore, we have included the following content: 

“ … In this review, we have reflected the literature most related to our topic, although we have not found previous analyses that explain the linear decline in alcohol consumption in France, from 23.2 in 1970 to 10.5 in 2021. …”. (Page 6).

4.) The background literature provided for USA is too short to support the implications given in the conclusion comments regarding concerns on the permanency of shocks and the need for strong measures to recover their original trends. More detailed information could be useful.

Reply: Yes, it is brief. Thank you for pointing that out. We have included the following literature and content: 

“… For the USA, the majority of the literature relates alcohol consumption to its impact on consumer mortality, highlighting it as a major concern. We found very few studies on the historical consumption patterns in this country. We reference the following: Grossman et al. [27] summarised (…). Additionally, Ritchie and Roser (2024) present data on alcohol consumption in the US, showing an increase from 2000 to 2019, consistent with the data presented in this analysis. …”. (Page 7). 

New reference:

Ritchie H, Roser M. Alcohol Consumption. Who consumes the most alcohol? How has consumption changed over time? And what are the health impacts? Our World in Data [Internet]. 2024 Jan [Cited 2024 Sept 10]. Available from: https://ourworldindata.org/alcohol-consumption.

5.) The concrete findings from each country are expected to be discussed in the light of existing challenges and the risk factors or preventing measures which are specific for the country.

Reply: Thank you for pointing out the need to contextualize the findings based on the specific challenges and preventive measures for each country. In response, we have revised the discussion section to incorporate a detailed analysis for each G7 country, addressing how the results relate to their individual challenges and policy contexts. For instance, we discuss how France’s long-standing alcohol culture has been mitigated by public health measures, whereas in Japan, there is a need for stronger regulations to address the rising alcohol consumption among women. Similarly, countries like the US and Canada, where persistence in alcohol consumption is high, require more robust regulatory interventions compared to countries like France and Japan, where mean reversion is evident. These discussions highlight the varying levels of persistence and the country-specific risks, ultimately providing a tailored understanding of the results. Pages 18-20.

6.) Although the data analyses and applied model seems sound and well-explained, the methodological limitations of such mathematical modelling are not included in the paper.

Reply: Thanks for the comment. We have included some additional comments on the limitations of the present approach. In fact, we mention in the new version alternative modelling frameworks producing very similar results.

7.) The manuscript is well-written; however, it is organized in a different way than it usually must appear (does not include discussion, strengths and limitations).

Reply: Thanks. We have added a new section for Discussion. Limitations of the present manuscript are included in the final section.

8.) The paper would be improved by relating the proper background information from each country to the findings and discussing the results in the light of the specific situations for the country. Thus, the implications and conclusion comments should be balanced and adjusted accordingly, rather than reporting some general and already known policy recommendations.

Reply: We also appreciate the comment about the need to better integrate background information with the findings and provide more balanced implications. In response, the conclusions have been revised to reflect specific policy implications for each G7 country, moving away from generalized recommendations. Associated with point 5, in countries with high persistence, such as the US, Canada, and Germany, we recommend sustained policy interventions, including taxation and regulatory measures. Meanwhile, for countries like Japan and France, where mean reversal is more evident, we emphasize the need to reinforce and monitor existing preventive measures. This approach ensures that the policy implications are directly related to the country-specific findings and tailored to address their unique contexts. Pages 21-23.

Other minor changes:

Page 24, Acknowledges: Comments from the Editor and two anonymous reviewers are gratefully acknowledged.

We have corrected several typos and grammatical errors in the new version of the manuscript.

References

Ritchie H, Roser M. Alcohol Consumption. Who consumes the most alcohol? How has consumption changed over time? And what are the health impacts? Our World in Data [Internet]. 2024 Jan [Cited 2024 Sept 10]. Available from: https://ourworldindata.org/alcohol-consumption.

Comments from Reviewer #2: 

The manuscript covers analysis of alcohol consumption trends in G7 countries during the very long period (1960-2021). To my opinion the manuscript is written by authors very well using using simple way of presentation of the data and understandable for each reader. 

I just have few technical remarks for the authors of the manuscript:

1.) Each abbreviation should be described at first use in the text (for example OECD). Such abbreviation first time used in page 5 of the manuscript but described only in page 10 of the manuscript. Please describe the meaning of the abbreviations SMI, CPI, and GDP used in Table 1 (for example, in the text at the bottom of the Table1.

Reply: Thanks for the comment. We have done so.

2.) Please modify title of the Table 2. For example, Descriptive statistics of alcohol consumption in G7 countries (in litres/capita) or so on.

Reply: Done. 

3.) Please indicate and explain what mean bolded values in Tables 3, 4 and 5.

Reply: Done. 

My recommendation is accept the manuscript after minor revision.

Reply: Many thanks for all your positive comments!

Other minor changes:

Page 24, Acknowledges: Comments from the Editor and two anonymous reviewers are gratefully acknowledged.

We have corrected several typos and grammatical errors in the new version of the manuscript.

---

## [Decision Letter · Decision Letter 1]

19 Nov 2024

Alcohol Consumption in the G7 Countries (1960-2021). Permanent versus Transitory Shocks.

PONE-D-24-04353R1

Dear Dr. Gil-Alana,

We’re pleased to inform you that your manuscript has been judged scientifically suitable for publication and will be formally accepted for publication once it meets all outstanding technical requirements.

Kind regards,

Ricardas Radisauskas

Academic Editor

PLOS ONE

Additional Editor Comments (optional):

Reviewers' comments:

Reviewer's Responses to Questions

**Comments to the Author**

1. If the authors have adequately addressed your comments raised in a previous round of review and you feel that this manuscript is now acceptable for publication, you may indicate that here to bypass the “Comments to the Author” section, enter your conflict of interest statement in the “Confidential to Editor” section, and submit your "Accept" recommendation.

Reviewer #1: All comments have been addressed

2. Is the manuscript technically sound, and do the data support the conclusions?

Reviewer #1: Yes

3. Has the statistical analysis been performed appropriately and rigorously? 

Reviewer #1: Yes

4. Have the authors made all data underlying the findings in their manuscript fully available?

Reviewer #1: Yes

5. Is the manuscript presented in an intelligible fashion and written in standard English?

Reviewer #1: Yes

6. Review Comments to the Author

Reviewer #1: Thank you for the constructive response letter and the revised manuscript. All the comments are sufficiently addressed and the revisions undertaken have improved the paper. Based on this, I recommend the editors to accept the manuscript for publication in PLOS ONE.

Good luck!

7. PLOS authors have the option to publish the peer review history of their article (what does this mean?). If published, this will include your full peer review and any attached files.

Reviewer #1: No

---

## [Editor Report · Acceptance letter]

21 Nov 2024

PONE-D-24-04353R1 

PLOS ONE

Dear Dr. Gil-Alana, 

I'm pleased to inform you that your manuscript has been deemed suitable for publication in PLOS ONE. Congratulations! Your manuscript is now being handed over to our production team.

Kind regards, 

on behalf of

Professor Ricardas Radisauskas 

Academic Editor

PLOS ONE